# HTR1A, TPH2, and 5-HTTLPR Polymorphisms and Their Impact on the Severity of Depressive Symptoms and on the Concentration of Tryptophan Catabolites during Hepatitis C Treatment with Pegylated Interferon-α2a and Oral Ribavirin (PEG-IFN-α2a/RBV)

**DOI:** 10.3390/cells12060970

**Published:** 2023-03-22

**Authors:** Tomasz Pawlowski, Krzysztof Malyszczak, Dariusz Pawlak, Małgorzata Inglot, Małgorzata Zalewska, Anna Grzywacz, Marek Radkowski, Tomasz Laskus, Justyna Janocha-Litwin, Dorota Frydecka

**Affiliations:** 1Department of Psychiatry, Wrocław Medical University, 50-367 Wrocław, Poland; 2Department of Pharmacodynamics, Medical University of Bialystok, 15-089 Białystok, Poland; 3Department of Infectious Diseases, Liver Diseases and Acquired Immune Deficiency, Wroclaw Medical University, 50-367 Wrocław, Poland; 4Independent Laboratory of Health Promotion, Pomeranian Medical University in Szczecin, 70-111 Szczecin, Poland; 5Department of Immunopathology of Infectious and Parasitic Diseases, Medical University of Warsaw, 02-091 Warsaw, Poland; 6Department of Adult Infectious Diseases, Medical University of Warsaw, 02-091 Warsaw, Poland; 7Department of Infectious Diseases and Hepatology, Wroclaw Medical University, 50-367 Wroclaw, Poland

**Keywords:** HCV, depression, pegylated interferon-α2a, HTR1A, TPH2, 5-HTT, TRYCATs

## Abstract

Background: Seeing that there are no data about associations between serotonin gene polymorphism and tryptophan catabolite concentration during PEG-IFN-α2a treatment, the aim of the current study is to examine (a) the associations between polymorphisms within the HTR1A, TPH2, and 5-HTT genes and the severity of depression symptoms and (b) the relationships among rs6295, rs4570625, and 5-HTTLPR rs25531polymorphisms and indoleamine 2,3-dioxygenase (IDO) activity, as well as kynurenine (KYN), tryptophan (TRP), kynurenic acid (KA), and anthranilic acid (AA) concentrations. Materials and methods: The study followed a prospective, longitudinal, single-center cohort design. The severity of the depressive symptoms of 101 adult patients with chronic HCV infections was measured during PEG-IFN-α2a/RBV treatment. We used the Montgomery–Åsberg Depression Rating Scale (MADRS) to assess the severity of depressive symptoms. The subjects were evaluated six times—at baseline and at weeks 2, 4, 8, 12, and 24. At all the time points, MADRS score, as well as KYN, TRP, KA, and AA concentrations, and IDO activity were measured. At baseline, rs6295, rs4570625, and 5-HTTLPR rs25531polymorphisms were assessed. Results: Subjects with C/C genotypes of 5-HT1A and lower-expressing alleles (S/S, L_G_/L_G_, and S/L_G_) of 5-HTTLPR scored the highest total MADRS scores and recorded the highest increase in MADRS scores during treatment. We found associations between TRP concentrations and the TPH-2 and 5-HTTLPR rs25531 genotypes. Conclusions: Our findings provide new data that we believe can help better understand infection-induced depression as a distinct type of depression.

## 1. Introduction

Understanding the etiology and pathogenesis of depression is of great importance to specialists and patients, as it is the leading cause of disability around the world [1]. Its nature remains an object of debate. One of the reasons is the fact that the diagnostic definition of depression is characterized by a heterogeneity caused by several different patterns of symptoms [2]. Some of the symptoms are quite contradictory, for example insomnia and hypersomnia or weight loss and gain.

According to Rantala et al., the subtype of the depressive episode determines which symptoms manifest [3]. Based on the processes and purposes of depressive episodes, they proposed an evolutionary subtyping of depression in their review. Based on evolutionary psychiatry, they divided depression into 12 subtypes: depression caused by infection, depression caused by long-term stress (previously melancholy), depression caused by loneliness, depression caused by traumatic events, depression caused by grief, depression caused by romantic rejection, postpartum depression, season-related depression, chemically induced depression, depression caused by somatic diseases, and depression caused by starvation.

For years, pegylated interferon-alfa (PEG-IFN-α) with ribavirin (RBV) have been the recommended line of treatment for chronic hepatitis C virus (HCV) infection. Nowadays, the standards of treatment for chronic HCV infection are direct-acting antiviral agents (DAAs) because they are safer and more effective than PEG-IFN-α [4]. DAAs can reduce neuroinflammation and foster white matter tract recovery [5,6]. This notwithstanding, IFN-α remains of interest to psychiatry because it allows for the prospective observation of the occurrence of infection-induced depression. One in four patients with chronic HCV who starts INT-α and RBV treatment develops a major depressive episode [7]. This type of depression is connected with the induction of indoleamine 2,3-dioxygenase (IDO), which leads to lower plasma tryptophan (TRP) and an increased synthesis of detrimental tryptophan catabolites (TRYCATs) [8]. In our previous study, we demonstrated that the activation of IDO connected with PEG-IFN-α2a/RBV treatment was prolonged, even 6 months after the end of treatment [9]. The induction of IDO was associated with a tryptophan depletion mechanism and neurodegeneration caused by TRYCATs—the neurodegeneration hypothesis [10]. We presented data according to which different mechanisms acquired particular significance during PEG-IFN-α2a/RBV HCV treatment in different time periods. In week 12 of treatment, a “depletion mechanism” in the form of tryptophan availability to the brain was crucial for the diagnosis of a major depressive disorder (MDD), whereas in week 24 of treatment, it was increased anthranilic acid concentrations—one of the TRYCATs—that had an impact on the incidence of MDD [11]. Overproduction of TRYCATs triggered immune activation [12]. Correlations were also found between the activation of peripheral blood mononuclear cells and the severity of depressive symptoms in patients with chronic hepatitis C [13].

Two recent large-scale meta-analyses by Almulla et al. have provided evidence that infection-induced depression can be a distinct type of depression based on its pathomechanism [14,15]. The first analysis showed that patients with MDD and BP were connected, with a decrease in TRP concentration without overactivity of the kynurenine pathway [9]. In a second study, Almull et al. found that patients with affective disorders with melancholic and psychotic features showed reduced availability of TRP to the brain with normal IDO activity [10].

The question that arises considers the possible role of vulnerability factors in moderating the effects of the inflammatory process. One of the important factors is genetic polymorphism, which can increase individual vulnerability to depression. Some data show that it is not psychological factors but depressive vulnerability that is associated with the severity of depressive symptoms among HCV patients treated with IFN-α [16].

In light of this, we decide to investigate the possible role of serotonergic system gene polymorphisms in the depletion of tryptophan or in the increase of serotonin reuptake by its transporter, leading to relative serotonin deficiency. Following Kraus et al. [17] and Lotrich [18], we select three genes and four polymorphisms:
The 5-HT1A receptor (HTR1A) gene located on chromosome 5: the major autoreceptor on the serotonergic raphe neurons, with a C-1019 G single-nucleotide polymorphism (rs6295).The tryptophan hydroxylaze-2 (TPH2) gene located on chromosome 12: a rate-limiting enzyme in the biosynthesis of serotonin in the brain, with a G-703 T single-nucleotide polymorphism (rs4570625).The serotonin transporter (5-HTT, SLC6A4) gene located on chromosome 17 with an insertion-deletion polymorphism of the 5-HTT gene transcription initiation site (5-HTTLPR), which produces short (S) alleles with lower transcriptional efficiency than long (L) alleles; in addition, a single-nucleotide polymorphism is investigated within the 5-HTTLPR A>G (rs25531) results in two variants of both the L allele (L_A_ or L_G_) and the S allele (S_A_ or S_G_). The L_G_ allele functions similarly to the S allele, as both of these demonstrate reduced transcriptional efficiency. In effect, 5-HTTLPR functionally is tri-allelic: L_A_, L_G_, and S [19].

Previous investigations among genes (HTR1A, TPH2, and 5-HTTLPR) involved in the serotoninergic system and their role in the increase in depressive symptoms during IFN treatment have attempted to determine the part of HTR1A and 5-HTTLPR polymorphisms. However, the data for HTR1A are inconsistent. Kraus et al. [11] found that homozygosity for the G allele predisposed patients to depression. In a study by Cozzolongo et al. [20], G-allele-carrier patients experienced greater depressive symptoms before and during IFN treatment. On the other hand, in a study by Almeida et al. [21], diagnosis of IFN-α-related depression was associated with the CC genotype.

The data for 5-HTTLPR are more consistent. Bull et al. found that L/L subjects developed fewer depressive symptoms in comparison to the S/S and L/S groups [22] during IFN treatment. Lotrich et al. evaluated the 5-HTTLPR single-nucleotide polymorphism and found that the L_A_ allele was associated with a lower rate of depression. The highest rate of depression was found among patients with S/S and S/L_G_ genotypes [12]. Another tri-allele study of 5-HTTLPR found that ethnicity could modulate the role of 5-HTTLPR as a vulnerability factor. Pierrucci-Lagha et al. found that, in a non-Hispanic Caucasian sample, L_A_/L_A_ homozygotes showed greater depressive symptoms, whereas among a Hispanic sample, the S/S, L_G_/L_G_, and S/L_G_ groups reported the highest rates of depression symptoms during IFN treatment. Analysis of the above-mentioned studies leads to an interesting observation: in studies that simultaneously investigated the impact of HTR1A and 5-HTTLPR polymorphisms on depressive symptoms, only HTR1A significantly influenced the increase in depressive symptoms or the diagnosis of depression [12,14,15].

Hence, the current study is conducted to examine:
The associations between polymorphisms within the HTR1A, TPH2, and 5-HTT genes and the severity of depression symptoms based on the Montgomery–Åsberg Depression Rating Scale (MADRS), a clinician-rated measure, during PEG-IFN-α2a/RBV treatment ([23]).The relationships among the above introduced polymorphisms and IDO activity, as well as kynurenine, tryptophan, kynurenic acid, and anthranilic acid concentrations, during PEG-IFN-α2a/RBV treatment.

To the best of our knowledge, this relationship has not been investigated elsewhere.

## 2. Materials and Methods

### 2.1. Participants

We enrolled 101 adult patients with chronic HCV infections with compensated liver disease. The subjects were scheduled to undergo therapy with PEG-IFN-α2a at 180 μg once weekly (Pagasys; Hoffmann—LaRoche, Basel, Switzerland) and RBV (Rebetol, Schering-Plough) in a dose of 1000 mg/day if the patient’s body weight was <75 kg or 1200 mg/day if a patient’s body weight was ≥75 kg for a duration of at least 24 weeks.

During the 24 weeks of treatment, 4 patients had to discontinue the treatment early because of PEG-IFN-α2a somatic side effects. The data of 97 patients who completed the 24 weeks of treatment were included in the analysis. The 97 subjects (48 men and 49 women; mean age and SD of 46.2 ± 9.9 years) were recruited from the same geographic area and belonged to native Polish populations. The participants in the current study were also a subsample of participants from previous studies analyzing alterations in the metabolism of tryptophan after interferon treatment and the role of kynurenine metabolites in the increase in depressive symptoms [9,11]. The detailed baseline characteristics of the study cohort are presented in Table 1.

Informed consent was obtained from each patient in the study, and the study protocol followed the ethical guidelines of the 1975 Helsinki Declaration. Furthermore, the study was approved by the ethical committee at the Wroclaw Medical University.

### 2.2. Study Design

The study followed a prospective, longitudinal, single-center cohort design. The subjects filled in a questionnaire regarding demography and history, which was followed by a baseline psychiatric evaluation using the Present State Examination from the Schedules of Clinical Assessment in Neuropsychiatry (SCAN 2.0) [24].

Exclusion criteria included clinical conditions (other than from HCV) that might influenced their immune status (coinfections, pregnancy, and autoimmune and neoplastic diseases), previous treatment with interferon α, a history of traumatic brain injury, and psychiatric disorders at baseline, such as MDD, bipolar disorder, or schizophrenia. The patients were not allowed to actively use alcohol. We used the MADRS to assess the severity of depressive symptoms. The subjects were evaluated six times—at baseline (week 0) and at weeks 2, 4, 8, 12, and 24—during PEG-IFN-α2a/RBV treatment by a senior board-certified investigator.

### 2.3. Genotyping

Genomic DNA samples were obtained from peripheral white blood cells from whole frozen blood using QIAamp DNA Blood Mini Kits (Qiagen GmbH, Hilden, Germany).

Single-nucleotide polymorphisms in the *TPH2* gene -703G>T (rs4570625) and *5HTR1A* gene (rs6295) were genotyped using validated and predesigned TaqMan^®^SNP Genotyping Assays (C_226207_10 and C_11904666_10). The polymorphism of 5’HTTLRP 3609 A>G (rs25531) was assessed with PCR RFLP using an *Msp*I FAST restriction enzyme. The PCR amplification was carried out with the 5’ GGCGTTGCCGCTCTGAATGC 3’ and 5’ GAGGGACTGAGCTGGACAACCAC 3’ primers in a total volume of 10 µl of solution containing 30 ng of genomic DNA. The polymorphism of 44bp ins/del (L/S) in the *5-HTT* gene was genotyped through PCR with a pair of primers (5’-TAMRA-GGC GTT GCC GCT CTG AAT GC-3’ and 5’GAG GGA CTG AGC TGG ACA ACC AC) using GeneAmp 970 PCR kits (Applied Biosystem, Waltham, MA, USA), followed by capillary electrophoresis using a DNA 3500 ABI sequenator (Applied Biosystem, USA). Individual genotypes were labeled according to their peak size with GeneMapper^®^ Software version 4.1, being 44 bp ins (long, L) (528 bp) and 44 bp del (short, S) (484 bp).

### 2.4. Blood Tryptophan, Kynurenine, and Anthranilic Acid Assessments

Tryptophan (TRP) and its metabolites were determined by high-performance liquid chromatography (HPLC). The chromatographic equipment was an Agilent Technologies 1260 Infinity series LC system composed of a G1321 binary pump (VL), a G1379B degasser (u-D), a G1329A autosampler (ALS), a G1330B thermostat for the autosampler, a G1316A column thermostat, a G1315C diode array (DAD VL+), and G7121B fluorescence detectors (FLDs).

Deproteinized samples were prepared by adding 20 μL of 2M perchloric acid into 100 μL of plasma. The acidified samples were vortexed, kept at 4 °C for 2 min, and then centrifuged for 30 min at 14,000× *g* at 4 °C. An amount of 2 μL of the supernatant was injected into the HPLC system for analysis. The prepared sample was separated on a Reprospher 100 C18 3.5 μm 2.1 × 150 mm column. The column effluent was monitored with a diode array detector for kynurenine (KYN) at 365 nm and tryptophan (TRP) at 280 nm. The mobile phase was composed of 0.1 M acetic acid and 0.1 M ammonium acetate (pH 4.6) containing 0.1% acetonitrile, and it was pumped at a flow rate of 0.2 mL/min. Chromatography was carried out at 24 °C.

Kynurenic acid (KYNA) and anthranilic acid (AA) concentrations were determined using a column (Phenomenex PEPTIDE 3.6 μm XB-C18 4.6 × 250 mm), and effluent was monitored using a programmable fluorescence detector. The optimized conditions were determined by recording fluorescence spectra with a stop-flow technique. Excitation and emission wavelengths were set at 254 and 404 nm, respectively. The mobile phase consisted of 100 mM of sodium acetate and 45 mM of zinc acetate containing 16% acetonitrile and was pumped at a flow rate of 0.6 mL/min. An amount of 2 μL of the supernatant was injected into the HPLC system for analysis. The output of the detector was connected to a single LC ChemStation instrument. Chromatography was carried out at 24 °C.

Chromatograms of each analyte are included in Appendix A.

### 2.5. Statistical Analysis

Evaluation of the Hardy–Weinberg equilibrium (HWE) was performed by comparing the observed and expected genotype distributions using the χ^2^ goodness-of-fit test. Distributions of alleles and genotypes of the 5-HT1A gene, TPH2 gene, and 5-HTT gene polymorphisms were assessed using the χ^2^ test. Differences were considered as statistically significant if the two-tailed *p*-value was less than 0.05.

Multiple repeated-measure regression models were fitted using a generalized estimating equation population-averaged models (Xtreg procedure in Stata program). In the generalized estimating equation models, we used an exchangeable covariance structure. The effects of explanatory variables on dependent variables were expressed using regression coefficients (β) and 95% confidence intervals (CIs). First, we used 3 separate models—total MADRS scores for 5-HT1A, TPH-2, and 5-HTTLPR rs25531 polymorphisms—with time serving as an explanatory variable.

Then, we used 15 (5 × 3) separate models for the kynurenine, tryptophan, kynurenic acid, and anthranilic acid concentrations and IDO activity, as well as for 5-HT1A, TPH-2, and 5-HTTLPR polymorphisms, and we used time as an explanatory variable.

The polymorphism within the L allele (L_G_) in the serotonin transporter length promotor region was functionally comparable with the S allele because of lower transcription efficiency [25].

For the statistical analyses, L_G_ and S alleles were grouped as the lower-expressing alleles and compared with higher-expressing L_A._ In the statistical analyses, we used 3 groups: L_A_/L_A_ (N = 34); S/L_A_ and L_G_/L_A_ (N = 43); and S/S, L_G_/L_G_, and S/L_G_ (N = 20).

We estimated the IDO enzymatic activity for each time point by calculating the kynurenine–tryptophan ratio × 10^3^ (mmol/mol).

## 3. Results

### 3.1. Distribution of Genotypes

The 5-HTR1A, TPH-2, and 5-HTTLPR rs2553 polymorphism frequencies are presented in Table 2. The frequency of distributions of all the polymorphisms was determined in accordance with Hardy–Weinberg equilibrium.

### 3.2. MADRS Score and KYN, TRP, KA, and AA Concentrations during PEG-IFN-α2a/RBV Treatment

Total MADRS score, KYN, KA and AA concentrations were significantly increased compared to baseline whereas TRP concentration decreased during PEG-IFN-α2a/RBV. (Table 3).

### 3.3. Serotonergic System Gene Polymorphisms and Their Relationships with MADRS Score

We found a main effect of 5-HT1A and 5-HTTLPR rs25531 genotypes but not the TPH-2 genotype on the severity of depressive symptoms, as presented in Figure 1, Figure 2 and Figure 3 and Table 4.

Concerning the 5-HT1A gene, the MADRS score significantly increased and differed between the C/C, G/G and C/G genotypes. Subjects with C/C alleles scored the highest MADRS points (above 18 points at the 8th and 12th weeks of treatment), and this group also recorded the highest increase in MADRS scores (between baseline and 12th week, it was 13.25 points). The lowest increase in MADRS score was observed among patients with G/G alleles.

The MADRS scores among the 5-HTTLPR genotypes significantly differed at the observed time points. Subjects with lower-expressing alleles (S/S, L_G_/L_G_, and S/L_G_) reached the highest MADRS scores (above 17 points at week 4). This group also experienced the highest increase in MADRS scores (between baseline and 4th week, it was 12.5 points). The group of patients with heterozygosity (S/L_A_ and L_G_/L_A_) had the lowest increase in MADRS score.

### 3.4. Serotonergic System Gene Polymorphisms and Their Relationships with Blood Tryptophan Metabolites

An analysis of the GEE models for kynurenine, tryptophan, kynurenic acid, and anthranilic acid concentrations, as well as IDO activity, for different serotonin gene polymorphisms revealed associations between tryptophan concentration and the TPH-2 and 5-HTTLPR rs25531 genotypes, as presented in Figure 4 and Figure 5 and Table 5.

The T-carrier genotype TPH-2 gene was associated with a significant decrease in tryptophan concentration over time. Furthermore, in the 12th week of treatment for the group of patients with homozygosity (T/T), we observed the highest decline in tryptophan concentration in comparison to baseline.

The tryptophan concentrations among the 5-HTTLPR genotypes significantly differed at the observed time points. The highest decline in tryptophan concentration in comparison to baseline was observed in the group with the lower-expressing alleles (S/S, L_G_/L_G_, and S/L_G_).

## 4. Discussion

### 4.1. Serotonergic System Gene Polymorphisms and Their Relationships with MADRS Score

This is the first study that reveals an association between the severity of depressive symptoms and polymorphisms within two serotonergic genes simultaneously in a cohort of patients treated with INT-α. We found that, among the tested serotonergic system genes, HT1A and 5-HTTLPR were involved in the increase in the severity of depressive symptoms.

Homozygosity for the C allele in the HTR1A gene scored the highest MADRS points (above 18 points) and showed the highest increase in MADRS scores in comparison with the G/G and C/G genotype subjects. Our findings corroborate those from a study by Galvao-de Almeida et al. in which the diagnosis of current depression demonstrated an association with the C/C genotype of the HTR1A gene [21]. Our results, however, disagree with those obtained by Kraus et al. [17] and Cozzolongo [20] et al. We believe that one of the reasons for the discrepancies might be methodological issues. Kraus et al. found that homozygosity for the G allele in the HTR1A gene significantly increased the incidence and severity of interferon-induced depression [17]. As their main psychometric instrument, they used the Hospital Anxiety and Depression Scale (HADS)—a self-assessment tool—and to assess the severity of depressive symptoms, they analyzed data from the seven-item HADS depression subscale, which they set to a ≥9 cut-off value for depression. Bowling stated that researchers who used questionnaires need to be aware of potential biasing effects on their data, and using a self-reporting instrument is connected with a danger of exaggeration or minimization by the responding patient [26]. According to Moore and Moore, the scores of patients with comorbid physical conditions may be falsely high if physical complaints such as fatigue suggest physical disease rather than depression [27]. For this reason, in our study we decided to use a clinician-rated measure for evaluation. We decided to use the MADRS rather than the Hamilton Depression Rating Scale (HAMD) because the MADRS was designed with more sensitivity to research responses generated by changes brought about due to antidepressants and other psychopharmacological interventions beyond what the HAMD proposes [23].

Setting the cut-off value for depression also means confronting potential bias because, as one of the important findings from the STAR*D study makes clear, using sum-scores as a proxy for depression severity may be unjustified [2]. According to Fried and Nesse, the assumption that an individual’s depression is adequately described by a sum-score may conceal important clinical insights [2].

It is worth noting that Galvao-de Almeida et al. [21], in a study whose results are consistent with ours, used a structured psychiatric interview (the Mini International Neuropsychiatric Interview) as a tool for diagnosing depression.

According to the findings of Cozzolongo et al. [20], patients carrying the G allele in the HTR1A gene experienced greater depressive symptoms before and during IFN treatment, which corroborates the results of Kraus’s study [17] but is contrary to what we established. Cozzolongo et al. used the same methodology for assessing depression severity and for diagnosing depression, i.e., the seven-item HADS depression subscale, and cut-off value for depression, but an additional methodological issue arose. Cozzolongo et al. discovered that the changes in depression scores were significantly higher at T1 and T2, with T1 in that study occurring after 12 weeks of treatment and T2 at the end of treatment. This is where the problem with the T2 assessment point emerges. Depression scores at T2 contained depression scores at the 24th week of treatment for patients with HCV genotypes 2 and 3 and at the 48th week of treatment for patients with HCV genotypes 1 and 4, so the T2 assessment point was composed of assessments at both the 24th week and 48th week.

Our finding that homozygosity for the C allele in the HTR1A gene was associated with the severity of depression score during interferon treatment might suggest that infection-induced depression can actually be a distinct type of depression. The G(-1019) 5-HT1A allele has been associated with MDD and has been replicated in most [28,29,30,31,32] but not all studies [33]. The proposed course of action relates to an increase in 5-HT1A autoreceptor levels in depressed subjects, which leads to a reduction in 5-HT neurotransmission [34]. On the other hand, depressed patients with the C/C genotype showed better performances in a battery of nine cognitive tests [35]. Donaldson et al. found more mRNA produced from the C- versus the G-alleles of rs6295 in the prefrontal cortices of nonpsychiatric control subjects [36].

The second gene involved in the increase in the severity of depressive symptoms according to our study was 5-HTTLPR. Subjects with lower-expressing alleles (S/S, L_G_/L_G_, and S/L_G_) reached the highest MADRS scores and also had the highest increase in MADRS scores during the 24th week of INT-α treatment. These findings correspond to those of Bull et al. [22] and Lotrich et al. [18]. In a study by Bull et al., subjects with S/S and S/L genotypes showed a significant increase in depressive symptoms at both weeks 8 and 24 during INT-α treatment [22]. Lotrich et al. evaluated 5-HTTLPR SNP and found the highest rate of depression among subjects with S/S and S/L_G_ genotypes [12].

The role of 5-HTTLPR in the pathogenesis of MDD was first reported by Caspi et al. [37]. They found that individuals carrying either one or two copies of the S allele were more likely to develop a major depressive disorder in response to stress than individuals with homozygosity (L/L). However, the data are not consistent. Some studies have confirmed these findings [38,39] while others have not [40,41]. The same inconsistency can be found in meta-analytic studies. On the one hand, Bleys et al. [42] stated that their meta-analysis provided new evidence for the robustness of the interaction between stress and 5-HTTLPR in depression, but on the other hand, another meta-analysis published in the same year by Culverhouse et al. [43] found no proof of a strong interaction between stress and the 5-HTTLPR genotype thats would contribute to the development of depression. One of the latest meta-analyses about the interaction among 5-HTTLPR, stress, and depression also added a temporal dimension, which resulted in a three-way interaction: gene x environment x time [44]. This meta-analysis revealed that the 5-HTTLPR and stress interaction was a dynamic process, producing different effects at different time points. The study indirectly confirmed the stipulation that, while S-allele carriers correlate with a higher risk of depression, they also entail a greater capability for recovery.

### 4.2. Serotonergic System Gene Polymorphisms and Their Relationships with Blood Tryptophan Metabolites

The three-way interaction approach, i.e., gene x environment x time, gave us an opportunity to interpret our findings regarding serotonin gene polymorphisms and their relationships with blood tryptophan metabolites. To the best of our knowledge, ours is the first study that investigates associations between three serotonin gene polymorphisms and TRYCATs. We found that the highest decline in tryptophan concentration in comparison to baseline was observed in the group with the lower-expressing alleles (S/S, L_G_/L_G_, and S/L_G_) of 5-HTTLPR, which allowed us to closer inspect the gene x environment (stress) x time interaction. The environment–stress factor was connected with INT-α treatment, which inducts IDO activity and leads to lower plasma tryptophan and increased synthesis of TRYCATs. Therapy with interferon facilitates prospective observation above stress factors in a time interval that is strictly defined and controlled by treatment duration. Therefore, we can assume that the genetic polymorphism in the 5-HTTLPR gene was, at the same time, associated with an increase in the severity of depressive symptoms and with a decrease in tryptophan concentration. Our findings are consistent with a tryptophan depletion study by Neumeister et al. in which the S-allele of 5HTTLPR constituted a risk factor for the development of depression during tryptophan depletion in healthy women [45]. Interesting results were also yielded by a tryptophan depletion study among subjects with remitted MDD vs. controls (without personal history of psychiatric disorders) [46]. Among patients with remitted MDD, tryptophan depletion induced significant increases in the severity of depression scores for all the genotype groups, but they were more prominent in carriers of the LA/LA, S/LA, and LG/LA genotypes than in S/S, LG/LG, and S/LG carriers. In our study, the decrease in tryptophan concentration was caused by the activation of IDO, and the pattern of increase in the severity of depression was different: increases were more prominent in carriers of S/S, L_G_/L_G_, and S/L_G_ than of L_A_/L_A,_ and smallest increase was found among S/L_A_ and L_G_/L_A_ carriers. In our opinion, this may be another clue that infection-induced depression can be a distinct type of depression.

Tryptophan is an essential amino acid for the human organism. There are four TRP metabolic pathways: the synthesis of proteins, the synthesis of serotonin and melatonin, the synthesis of tryptamine, and the kynurenine pathway. The KYN pathway is responsible for almost 95% of dietary TRP degradation [47]. The first stage of the pathway consists of the metabolism of L-tryptophan into N-formylo-L-kynurenine, i.e., the first component in the kynurenine pathway. It can be catalyzed by two enzymes: tryptophan 2,3-dioxygenase (TDO) or indolamine 2,3-dioxygenases (IDO-1 and IDO-2). TDO expression is restricted to the liver in the periphery, and its activity is regulated by its substrate (tryptophan) and corticosteroids [48]. TDO activity is inhibited when IDO activity rises in other tissues of the body due to inflammatory reactions [49].

Although IDO-1 catalyzes the same biochemical reaction as TDO, in the case of IDO the substrates might include—apart from L-TRP—D-TRP, serotonin, melatonin, and tryptamine. IDO-1 occurs in extrahepatic tissues such as the kidneys, lungs, spleen, intestine, brain (especially the hypothalamus), placenta, epididymis, and endocrine glands, as well as in the monocytes of peripheral blood. In physiological conditions, the activity of IDO-1 is minimal, but it is highly induced by the following: interferons γ, α, and β; bacteria lipopolysaccharides; viruses; proinflammatory interleukins (IL-1, IL-6, and IL-8); and tumor cells [50]. A significant rise in the activity of IDO-1 results in a local (i.e., occurring in the affected tissue) fall in the levels of tryptophan. IDO-2 is present in tissues such as the epididymis, liver, and kidneys [51]. IDO-1 can act as a physiological regulator of immune system activation and is expressed in dendritic cells, secondary lymphoid organs, and pancreatic cells [52]. TRP metabolites are ligands for the aryl hydrocarbon receptor (AHR) [53]. The AHR can alter the inflammatory response on diverse cell types [54]. IFN-α, through inducing the activity of IDO-1, leads to decreases in the levels of blood plasma tryptophan and, as a consequence, those of serotonin. It was hypothesized that the induction of the kynurenine pathway may reduce the availability of TRP and, thus, lead to reduced serotonin synthesis and depression [8]. Decreased tryptophan availability to the brain has been detected in studies by Capuron et al. (cancer patients undergoing cytokine therapy) [55] and Pawlowski et al. (HCV patients undergoing PEG-IFN-α2a/RBV treatment) [11].

Our last finding was connected with the TPH-2 gene. The T-carrier genotype was associated with a significant decrease in tryptophan concentration over time. In comparison to baseline, at the 12th week of treatment the subjects with homozygosity (T/T) expressed the highest decline in tryptophan concentration. This is the first research report about the association between tryptophan concentration and TPH-2 gene polymorphism. Inoue et al. found that T-allele carriers were associated with significantly smaller volumes in the bilateral amygdala and hippocampus and a higher reward dependence than those with the G-allele homozygote [56]. A meta-analysis by Gao et al. reported an association between rs4570625 carrying the T-allele and MDD [57]. The results of a meta-analysis by Liu et al. also showed that rs4570625 may be significantly associated with depression [58].

## 5. Limitations

The main limitation of the current study was its open-label design. Given the limited number of therapies of PEG-IFN-α2a/RBV conducted at the center, patients could be overly optimistic about the impact of the studied interferon treatment. Participants were recruited from one center, so they cannot be seen as representative of the whole chronic HCV population. Since all the subjects came from a native Polish population, the interpretation and application of the findings of the present study naturally exclude other nations and ethnic groups. Lastly, because the sample size was relatively small, it precluded us from conducting further subgroup analyses.

## 6. Conclusions

Our findings indicated that homozygosity for the C allele in the HTR1A gene and the 5-HTTLPR S/S, LG/LG, and S/LG genotypes were associated with the severity of depressive symptoms during interferon treatment. These results could have clinical implications and could be applied to help identify subjects who are at an increased risk of IFN-α depression. Moreover, according to the results we obtained, genetic polymorphism in the 5-HTTLPR gene was associated with an increase in the severity of depressive symptoms and, at the same time, it was connected with decreased tryptophan concentration, which in turn gave insight into the gene x environment (stress) x time interaction when depressive symptoms arise. In our opinion, the above findings, together with the discovery that homozygosity (T/T) in the TPH-2 gene was associated with the highest decline in TRP concentration, may be another clue indicating that infection-induced depression can be a distinct type of depression.

## Figures and Tables

**Figure 1 cells-12-00970-f001:**
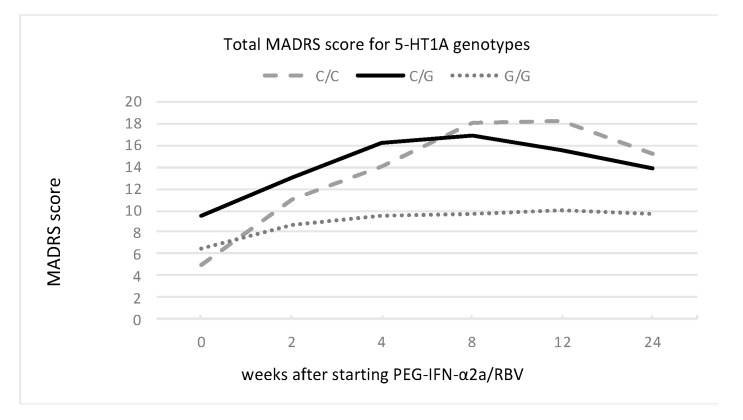
Changes in MADRS score for 5-HT1A genotypes of C/C, C/G, and G/G during PEG-IFN-α2a/RBV treatment.

**Figure 2 cells-12-00970-f002:**
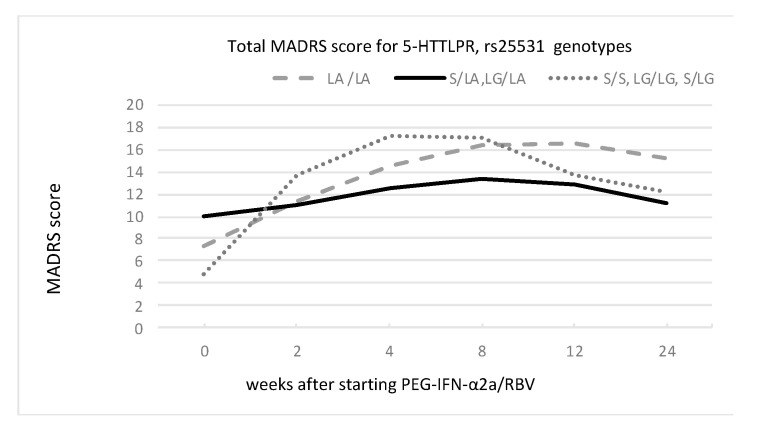
Changes in MADRS score for functional 5-HTTPLR genotype groups of L_A_/L_A_; S/L_A_ and L_G_/L_A_; and S/S, L_G_/L_G_, and S/L_G_ during PEG-IFN-α2a/RBV treatment.

**Figure 3 cells-12-00970-f003:**
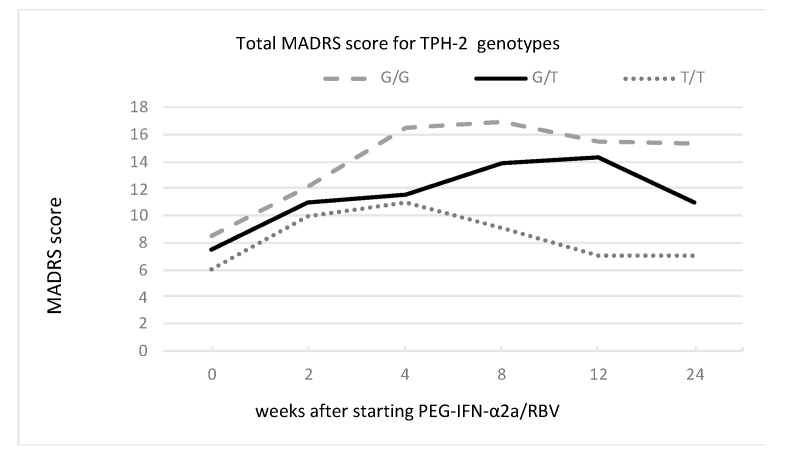
Changes in MADRS score for TPH-2 genotypes of G/G, G/T, and T/T during PEG-IFN-α2a/RBV treatment.

**Figure 4 cells-12-00970-f004:**
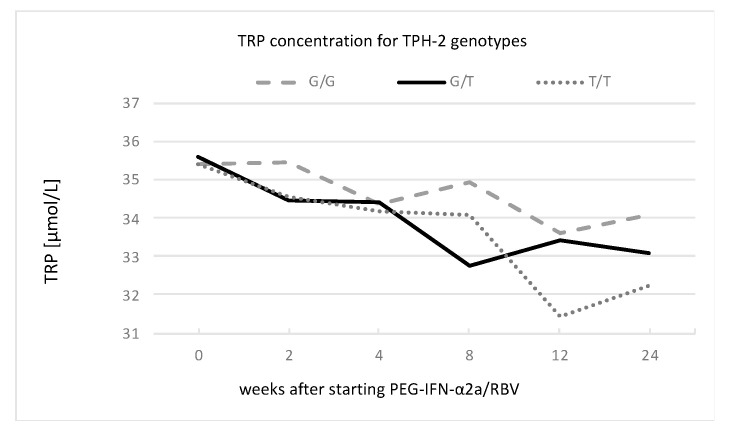
Changes in tryptophan concentration for TPH-2 genotypes of G/G, G/T, and T/T during PEG-IFN-α2a/RBV treatment.

**Figure 5 cells-12-00970-f005:**
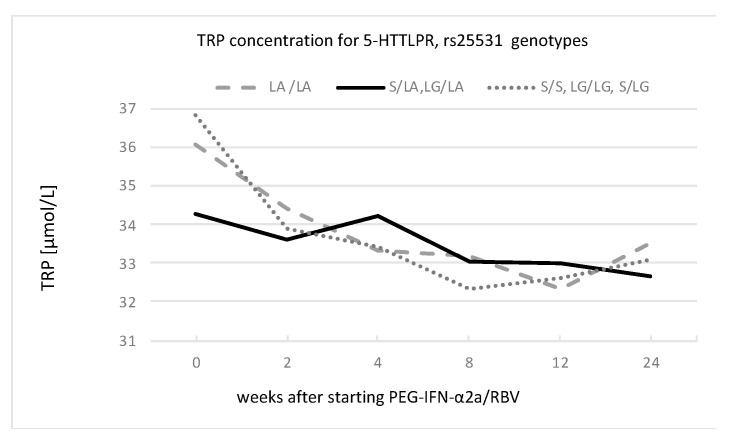
Changes in tryptophan concentration for functional 5-HTTPLR genotype groups of L_A_/L_A_; S/L_A_ and L_G_/L_A_; and S/S, L_G_/L_G_, and S/L_G_ during PEG-IFN-α2a/RBV treatment.

**Table 1 cells-12-00970-t001:** Sociodemographic characteristics of HCV patients treated with PEG-IFN-α2a/RBV (N = 97) (all values are given as mean ± SD).

Age (years)		46.2 ± 9.9
Male		48
Female		49
Weight (kg)		77.8 ± 16.3
Years of education		13 ± 3
Alanine aminotransferaze (IU/L)		83 ± 52
Serum HCV-RNA (×10^6^ IU/mL)		2.0 ± 3.3
HCV genotype	1	67
	3A	25
	4	5

**Table 2 cells-12-00970-t002:** Hardy–Weinberg’s analysis for cohort of patients.

Hardy–Weinberg Equilibrium Calculator		Observed	Allele Freq	Test χ^2^	
		(Expected)		χ^2^	*p*
5-HT1A rs6295	C/C	27 (26.29)	C = 52.06% G = 47.94%	0.08	0.77
	C/G	47 (48.42)			
	G/G	23 (22.29)			
TPH-2 rs4570625	G/G	51 (54.19)	G = 74.74% T = 25.26%	2.94	0.08
	G/T	43 (36.62)			
	T/T	3 (6.19)			
5-HTTLPR	L/L	40 (40.92)	L = 64.95% S = 35.05%	0.16	0.68
	L/S	46 (44.16)			
	SS	11 (11.92)			
functional 5-HTTPLR	L_A_/L_A_	34 (31.76)	L_a_ = 57.22% L_G_ = 7.73% S = 35.05%	6.72	0.081
	L_A_/L_G_	4 (8.58)			
	L_G_/L_G_	2 (0.58)			
	S/L_G_	7 (5.26)			
	S/S	11 (11.92)			
	S/L_A_	39 (38.91)			

*p*—statistical significance of χ^2^ test.

**Table 3 cells-12-00970-t003:** Means and standard deviations of MADRS score and KYN, TRP, KA, and AA concentrations in study cohort (N = 97).

	Week of Treatment					
	Baseline	2	4	8	12	24
Total MADRS score	7.0 (4.8)	11.3 (6.5)	13.8 (7.5)	14.7 (7.5)	14.1 (7.3)	14.7 (7.4)
KYN μmol/L	2.04 (0.66)	2.11 (0.57)	2.13 (0.69)	2.22 (0.64)	2.20 (0.61)	2.41 (0.60)
TRP μmol/L	35.1 (8.6)	34.2 (7.1)	33.7 (6.0)	33.1 (8.2)	32.3 (7.8)	32.5 (7.1)
KA nmol/L	27.6 (9.3)	26.2 (10.5)	27.5 (12.1)	30.5 (17.5)	31.5 (16.5)	34.0 (18.1)
AA nmol/L	44.6 (23.5)	52.9 (24.4)	59.0 (29.6)	60.7 (26.5)	49.9 (26.4)	54.8 (18.8)

**Table 4 cells-12-00970-t004:** Multilevel regression analyses indicating differences in MADRS score for each time point compared to baseline, with β indicating regression coefficient and *p* the level of significance for patients with genotypes of 5-HT1A, TPH-2, and functional 5-HTTPLR.

Weeks		2		4		8		12		24	
Genotype		Β	*p*-Value	Β	*p*-Value	β	*p*-Value	β	*p*-Value	β	*p*-Value
5-HT1A	C/C	6.00	<0.001	9.5	<0.001	13.00	<0.001	13.25	<0.001	10.25	<0.001
	C/G	−2.53	NS	−2.85	NS	−5.58	<0.05	−7.21	<0.01	−5.83	<0.05
	G/G	−3.83	NS	−6.5	<0.05	−9.83	0.01	−9.75	0.01	−7.08	<0.05
	L_A_/L_A_	4.09	<0.01	7.27	<0.001	9.08	<0.001	9.28	<0.001	7.98	<0.001
	S/L_A_, L_G_/L_A_	−3.29	NS	−5.02	<0.05	−5.94	<0.01	−6.58	0.001	−7.06	<0.001
	S/S, L_G_/L_G_, S/L_G_	4.90	NS	5.22	<0.05	3.16	<0.05	−0.28	NS	−0.48	NS
TPH-2	G/G	3.57	0.005	7.91	<0.001	8.30	<0.001	6.89	<0.001	6.72	<0.001
	G/T	−0.07	NS	−3.81	NS	−1.90	NS	−0.09	NS	−3.22	NS
	T/T	0.42	NS	−2.91	NS	−5.30	NS	−5.89	NS	−5.72	NS

**Table 5 cells-12-00970-t005:** Multilevel regression analyses indicating differences in tryptophan concentration for each time point compared to baseline, with β indicating regression coefficient and *p* the level of significance for patients with genotypes of TPH-2 and functional 5-HTTPLR.

Weeks		2		4		8		12		24	
Genotype		β	*p*-Value	β	*p*-Value	β	*p*-Value	Β	*p*-Value	Β	*p*-Value
5-TPH-2	G/G	0.01	<0.05	0.01	NS	0.05	NS	−0.07	NS	−0.02	NS
	G/T	−0.22	<0.01	−0.09	NS	−0.33	<0.001	−0.14	NS	−0.22	<0.01
	T/T	−0.19	NS	−0.03	NS	−0.11	NS	−0.31	<0.05	−0.10	NS
5-HTTPLR	L_A_/L_A_	−0.16	0.05	−0.27	0.001	−0.18	<0.05	−0.37	<0.001	−0.24	<0.01
	S/L_A_, L_G_/L_A_	−0.09	NS	−0.26	<0.05	−0.06	NS	−0.24	<0.05	7.06	NS
	S/S, L_G_/L_G_, S/L_G_	−0.13	<0.05	−0.07	NS	−0.26	<0.05	−0.04	NS	−0.10	NS

## Data Availability

The data presented in this study are available on request from the corresponding author. The data are not publicly available due to ethical restrictions.

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
