# Peer review of "HTR1A, TPH2, and 5-HTTLPR Polymorphisms and Their Impact on the Severity of Depressive Symptoms and on the Concentration of Tryptophan Catabolites during Hepatitis C Treatment with Pegylated Interferon-α2a and Oral Ribavirin (PEG-IFN-α2a/RBV)"

_cells, 2023, doi:10.3390/cells12060970_

Round 1

Reviewer 1 Report

Pawlowski et al. performed a study that aimed to investigate the potential implication of HTR1A, TPH2 and 5-HTTLPR polymorphisms in the relationship betwen depressive symtoms and tryptophan catabolites concentration with chronic hepatitis C patients following PEG-IFN-α2a/RBV treatment. Given the evidence suggesting that infection-induced depression has its own entity, this study address an interesting scientific cuestion that might help to improve our understanding of the mechanisms involved in this depresive phenotype. Despite that, there are a number of concerns that the authors should consider:

1. Four polymorphisms were included in the analyses, meaning that different comparisons were carried out. For this reason, a multiple test correction should be applied in order to correct for spurious associations.

2.  Althougth methods are written in detail, there is no mention about the software that was used for the analyses. This information may be of help.

3. The tables did not include the results for the 4 polymorphisms. It would be interesting to review all the results of the study. 

Minor comments:

1. English style should be reviewed throughout the manuscript. As an example, the verd in the following sentence is missing: The findings from Cozzolongo et al. [20] that G allele carrier patients in the HTR1A gene experienced greater depressive symptoms before and during IFN-treatment corroborate the results of Kraus’s study [17] but are contrary to what we established. 

2.  This reviewer did not find the information about: 1. Author Contributions; 2. Institutional Review Board Statement; 3. Informed Consent Statement. 4. Data Availability Statement; 5. Acknowledgments; and 6. Conflicts of Interest.

Reviewer 2 Report

The work of Pawloski et al. evaluates the role of serotonergic pathway gene polymorphisms (HTR1A, TPH2 and 5-HTTLPR), tryptophan (TRP) and TRYCATs, which also implies the IDO activity, in the severity of MDD symptoms during Hepatitis C Treatment with Pegylated Interferon-α2a and Oral Ribavirin (PEG-IFN-α2a/RBV). The authors' approach is interesting, as they make valuable contributions to this topic by integrating elements such as serotonergic pathway gene polymorphisms and TRP/TRYCAT concentrations in a multiple regression analysis; but there are some points that will help to improve the work:

comments:

1)      Please revise the wording of the title, it seems that tryptophan (TRP) and TRYCATs concentrations are not associated with MDD symptoms but only with IFN-treated HVC.

2)      2.1. Participants: please revise the sentence that describes the dose of IFN/RBV, although it is obvious, the text is not clear about what the RBV dose is.

3)      2.4 TRP/TRYCATs methods: references 25 and 26 are very old, it is important that the authors include a detailed description of the HPLC methods they performed; in addition, some typical chromatograms of each analyte should be included to enrich the work (or at least as supplementary figures). The paragraph should also be rewritten, it seems that the Holmes method is not by HPLC, but only the Hervé method.

4)      2.5 statistical analysis: please use the equation editor to obtain a proper chi-square, X is seen as a subscript of 2.

5)      Throughout the text the authors use the expression "serotonin gene polymorphisms"; I suggest using "serotonergic pathway gene polymorphisms" or "serotonergic system gene polymorphisms ".

6)      Lines 234 and 240: the authors say "more than 12 points and more than 13 points", respectively. I suggest using the exact value, since "more than" gives the idea that it is much more, and as we can see in the graph the values are very close to 12 and 13.

7)      The declarations section is empty, check and complete it.

8)      Line 157: the authors cite a table with detailed baseline characteristics of the study cohort from the previous paper; these data must be included and detailed in this paper.

9)      A table describing the mean and standard deviation of the MADRS, KYN, KA, TRP and AA score for each time point must be included.

10)   Discussion: the discussion could be enhanced by addressing in more detail the mechanism of tryptophan depletion by IFN-induced activation of the kynurenine pathway and the effects of this on the immune system and the central nervous system, as well as the relationship of this to the onset of depressive symptoms. This will provide a richer context for the results presented in this paper.

Round 2

Reviewer 1 Report

As stated in the previous revision, a multiple testing correction must be applied, given that this study includes the analysis of 4 independent polymorphisms. This is necessary to avoid false positive findings. 

Author Response

Dear reviewer

 Table 2 Hardy-Weinberg's analysis for cohort of patients shows how small numbers were obtained - e.g. TPH-2 T/T N=2, which meant that multiple testing correction could not be performed in this methodology. We agree with You that this is a weakness of our study, which is why we wrote about it in the Limitation section.

Regards

TP

Reviewer 2 Report

comments:

I continue to see serious deficiencies in the description of the methodology, especially in the HPLC section. In addition to the fact that the methodology is not congruent:

1)e.g. they use 20 mL of percloric acid to deproteinized “100 mL” of plasma?

R= First, they did not describe a plasma collection; second, did they really collect 100 ml of plasma from a person for this protocol?

2)The authors state that: “2ml of the supernatant was injected into the HPLC system”

R= Did they really inject 2 ml of sample into a column of dimensions 2.1x150 mm (Reprospher 100 C18 3.5 mm 2.1 x 150 mm)? A column with these characteristics has a column volume (p.r2.h) of about 500 microliters, and considering the packing volume, the dead volume is about 250ul. This means that the sample is 8 times the dead volume of the column; with these volumes, it is not possible to separate a peak at minutes 5, 6 or 7, as indicated by the authors in the chromatogram. At a flow rate of 0.2 ml/min, the 2 ml of sample needs ten minutes to fully load onto the column.

2.1)if an isocratic method was used, please describe the method of cleaning the column between injections.

3) it is necessary to improve the presentation of chromatograms.

4) Tables 1 and 3 must be improved.

My recommendation to the authors is that they take the time to carefully check all the details of the methodology, and that the methodology is correctly described so that other authors can reproduce their results. They must check the suitable format of tables and a suitable presentation of figures or images. I am sure that these points of the article can be improved considerably, however, the current version still has serious shortcomings.

Author Response

Dear Reviewer

First of all, I would like to apologize for the misunderstanding. The problem arose from the fact that the editorial system creates an article template that is not the original file we were working on.

My mistake was that the answer to the HPLC comments was prepared by our expert Prof. Pawlak and I pasted it into the article. Unfortunately, the Palatino editorial font changed some things, e.g. microlitres to ml, and I didn't notice. Currently, I applied the HPLC methodology, our Arial font, in the article template, but I have no idea if the editorial system will change anything. In addition, only 1 attachment can be placed in the response system for the reviewer, so the combination in one attachment together with the presentation of chromatograms text affects the quality.

Tables 1 and 3, after accepting their substantive value, will be changed to a specific format by the editors - as they did with the previous ones.

Round 3

Reviewer 2 Report

Thank you, I have no further comments.